# Regulation of Leaf Angle Protects Photosystem I under Fluctuating Light in Tobacco Young Leaves

**DOI:** 10.3390/cells11020252

**Published:** 2022-01-12

**Authors:** Zhi-Lan Zeng, Hu Sun, Xiao-Qian Wang, Shi-Bao Zhang, Wei Huang

**Affiliations:** 1Kunming Institute of Botany, Chinese Academy of Sciences, Kunming 650201, China; cengzhilan@mail.kib.ac.cn (Z.-L.Z.); sunhu19@mails.ucas.ac.cn (H.S.); wangxiaoqian@mail.kib.ac.cn (X.-Q.W.); sbzhang@mail.kib.ac.cn (S.-B.Z.); 2University of Chinese Academy of Sciences, Beijing 100049, China

**Keywords:** leaf angle, fluctuating light, photosynthesis, photoinhibition, photoprotection, photosystem I

## Abstract

Fluctuating light is a typical light condition in nature and can cause selective photodamage to photosystem I (PSI). The sensitivity of PSI to fluctuating light is influenced by the amplitude of low/high light intensity. Tobacco mature leaves are tended to be horizontal to maximize the light absorption and photosynthesis, but young leaves are usually vertical to diminish the light absorption. Therefore, we tested the hypothesis that such regulation of the leaf angle in young leaves might protect PSI against photoinhibition under fluctuating light. We found that, upon a sudden increase in illumination, PSI was over-reduced in extreme young leaves but was oxidized in mature leaves. After fluctuating light treatment, such PSI over-reduction aggravated PSI photoinhibition in young leaves. Furthermore, the leaf angle was tightly correlated to the extent of PSI photoinhibition induced by fluctuating light. Therefore, vertical young leaves are more susceptible to PSI photoinhibition than horizontal mature leaves when exposed to the same fluctuating light. In young leaves, the vertical leaf angle decreased the light absorption and thus lowered the amplitude of low/high light intensity. Therefore, the regulation of the leaf angle was found for the first time as an important strategy used by young leaves to protect PSI against photoinhibition under fluctuating light. To our knowledge, we show here new insight into the photoprotection for PSI under fluctuating light in nature.

## 1. Introduction

In nature, fluctuations of light intensity are typical light conditions experienced by leaves [1,2]. Upon a sudden increase in irradiance, the rapid increased electron flow from photosystem II (PSII) is accompanied by the relatively slower kinetics of diffusional conductance and CO_2_ fixation [3,4], leading to the transient PSI over-reduction [5,6,7,8]. Under such conditions, the electron donation from PSI to O_2_ increases, generating reactive oxygen species (ROS) within PSI [9]. Concomitantly, the antioxidant systems cannot immediately scavenge these ROS [10], making PSI damaged when exposed to fluctuating light (FL) [11,12,13,14]. PSI activity is characterized by a slow rate of recovery that needs several days [15,16,17]. The decrease in PSI activity suppresses CO_2_ fixation and thus impairs plant growth [16,18,19].

A key reason for PSI photoinhibition is PSI over-reduction, which is termed as PSI acceptor-side limitation [9,20,21,22]. The PSI redox state is controlled by donor and acceptor-side regulation [6]. Plants have several photoprotective mechanisms to protect PSI under FL [23,24,25]. In donor-side regulation, the downregulation of the electron flow from PSII or cytochrome b6f to PSI alleviates PSI over-reduction [22,26,27]. In acceptor-side regulation, the electrons in PSI can be consumed by electron sinks downstream, such as CO_2_ assimilation, photorespiration, water–water cycle and O_2_ photoreduction mediated by flavodiiron proteins [28,29]. In non-angiosperms, flavodiiron proteins mediate the photoreduction of O_2_ and thus prevent FL-induced PSI photoinhibition [30,31,32,33,34,35]. However, flavodiiron proteins are completely lacking in angiosperms [5]. Alternatively, the cyclic electron flow (CEF) around PSI is widely employed by angiosperms to fine-tune the PSI redox state under FL [6,9,36]. When the irradiance suddenly increases, CEF first increases to help the rapid formation of the proton gradient (∆pH) across the thylakoid membranes [37,38,39]. In general, a higher ∆pH not only downregulates plastoquinone oxidation at the cytochrome b6f complex but also increases ATP synthesis [40]. The impairment of CEF causes the loss of ∆pH formation and thus accelerates FL-induced PSI photoinhibition in *Arabidopsis thaliana* and rice (*Oryza sativa*) [6,9]. Consistently, the overexpression of CEF alleviates FL-induced PSI overreduction in C4 plant *Flaveria bidentis* [41].

In plants grown in open fields, the light intensity shining on leaves is largely determined by the leaf angle. If the leaf angle is horizontal, the light intensity shining on leaves is the maximum at noon. Concomitantly, the intensity difference between low and high light is also maximized on cloudy days. By comparison, if the leaf angle is vertical, the light intensity shining on leaves decreases at noon, and the intensity difference between low and high light is also minimized. Previous studies have indicated that the change in leaf angle can alleviate PSII photoinhibition under high light in beans (*Phaseolus vulgaris*) [42,43] and *Bauhinia tenuiflora* [44,45]. However, the role of the leaf angle in protecting PSI against photoinhibition under FL is little known. Recently, it has been indicated that FL-induced PSI photoinhibition is determined by the intensity difference between low and high light [46]. Therefore, the leaf angle can largely influence the risk of FL-induced PSI photoinhibition. However, no study has directly investigated the role of the leaf angle in safeguarding PSI under FL.

An interesting observation in tomatoes is that mature leaves tend to be horizontal, but the young leaves tend to be vertical. Young leaves have lower photosynthetic light use efficiency than mature leaves [14,47,48]. A constant high light can also induce significant photoinhibition of PSII in leaves with low photosynthetic capacity [49], leading to the impairment of plant growth. However, it is unclear whether this difference in leaf angle between tobacco mature and young leaves is related to photoprotection for PSI or PSII against constant high light.

In this study, we measured photosynthetic performances under constant high light and fluctuating light in mature and young leaves of tobacco seedlings. The main aims were to (1) determine whether PSI is more susceptible to FL in young leaves than in mature leaves and (2) assess the role of the leaf angle in the photoprotection of young leaves.

## 2. Materials and Methods

### 2.1. Plant Materials and Growth Condition

Tobacco (*Nicotiana tabacum* L. cv. K326) plants were grown in full sunlight with day/night air temperatures of 30 °C/20 °C and the maximum sunlight intensity at noon of 2000-μmol photons m^−2^ s^−1^. Plants were cultivated in plastic pots. The initial soil N content of humus soil is 2.1 mg/g. To avoid any nutrient stress, Peter’s Professional water solution (N:P:K = 15:4.8:24.1) was used for fertilizing once every two days. Furthermore, the plants were watered every day to avoid drought stress.

### 2.2. Experimental Design

All experiments were conducted in the morning (9:00–11:00 a.m.) on different clear days. Before the photosynthetic measurements, the leaf angles were measured. The leaf angle with a horizontal line was measured using a protractor, as shown in Figure 1. After dark adaptation for 20 min, the leaves were exposed to 1809-μmol photons m^−2^ s^−1^ for 20 min to measure the steady-state parameters. Afterwards, the leaves were exposed to FL alternating between 59- and 1809-μmol photons m^−2^ s^−1^ every 2 min for 24 min. Subsequently, the PSI activity was measured three times for each leaf to decrease the error, because the *P_m_* value might change slightly between independent measurements.

### 2.3. Chlorophyll Fluorescence and P700 Measurements

A Dual-PAM 100 measuring system (Heinz Walz, Effeltrich, German) was used to measure the PSI and PSII parameters for attached leaves under atmospheric CO_2_ condition [50]. The PSI parameters were calculated using the following equations: the quantum yield of PSI photochemistry, Y(I) = (*P_m_*′−*P*)/*P_m_*; the quantum yield of PSI nonphotochemical energy dissipation due to donor-side limitation, Y(ND) = *P*/*P_m_*; and the quantum yield of PSI nonphotochemical energy dissipation due to acceptor-side limitation, Y(NA) = (*P_m_*−*P_m_*′)/*P_m_*. The PSII parameters were calculated as follows: the effective quantum yield of PSII photochemistry, Y(II) = (*F_m_*′−*F_s_*)/*F_m_*′; the quantum yield of nonregulatory energy dissipation in PSII, Y(NO) = *F_s_*/*F_m_*; and the quantum yield of regulatory energy dissipation in PSII, Y(NPQ) = 1−Y(II)−Y(NO). The relative photosynthetic electron transport rates were calculated as follows: rETRI = PPFD × Y(I) × 0.84 × 0.5; rETRII = PPFD × Y(II) × 0.84 × 0.5. The activation of CEF was calculated by the ratio of rETRI/rETRII.

### 2.4. Statistical Analysis

The data were shown as the mean values of five individual experiments. Turkey’s multiple comparison test was used to determine the significant differences between different treatments (*α* = 0.05).

## 3. Results

### 3.1. Photoprotection under Constant High Light Is Not Affected by Leaf Angle

We first examined the photosynthetic performances under constant high light between the horizontal leaves (leaf angle = 0°) and vertical leaves (leaf angles 60–65°) (leaf angle measurements are indicated in Figure 1). After illumination at 1809-μmol photons m^−2^ s^−1^, we first examined the steady-state PSI and PSII performances in leaves with different leaf angles. The results indicated that the donor-side limitation (Y(ND)) and acceptor-side limitation (Y(NA)) of PSI were not affected by the leaf angle (Figure 2A). Y(ND) and Y(NA) reflected the PSI redox state that influenced the risk of PSI photoinhibition, and the performances of Y(ND) and Y(NA) indicated that PSI was insusceptible to constant high light. Furthermore, the quantum yields of the PSII regulatory energy dissipation (Y(NPQ)) and nonregulatory energy dissipation (Y(NO)) were not altered by the change in leaf angle (Figure 2B). Therefore, when exposed to constant high light, the excess light energy could be harmlessly dissipated via nonphotochemical quenching to diminish the risk of PSII photoinhibition.

### 3.2. Photosynthetic Responses to Fluctuating Light Is Affected by Leaf Angle

We next compared the photosynthetic performances under FL between the horizontal and vertical leaves. Under low and high light, Y(I), the quantum yield of PSI photochemistry, was higher in the horizontal leaves than the vertical leaves (Figure 3A). Upon transitioning to high light, the increase in Y(ND) was slower in the vertical leaves than the horizontal leaves (Figure 3B). As a result, within the first 10 s after transitioning to high light, the vertical leaves displayed higher Y(NA) than the horizontal leaves (Figure 3C). Therefore, the response of the PSI redox state to FL varied among leaves with different leaf angles. In the horizontal leaves, the low level of Y(NA) prevented the FL-induced photoinhibition of PSI (Figure 3C). In contrast, the relatively higher Y(NA) in the vertical leaves increased the risk of PSI photoinhibition under FL.

During FL treatment, the horizontal leaves had a higher quantum yield of PSII photochemistry (Y(II)) than the vertical leaves (Figure 4A). After transitioning to high light, the vertical leaves had slightly higher Y(NPQ) and similar Y(NO) values (Figure 4B,C). These results indicate that the PSII performance under FL hardly varied in leaves with different leaf angles. Therefore, for leaves with contrasting leaf angles, the effect of FL on the PSI performance was different from that on PSII. During the high light phases in FL, the horizontal leaves had higher rETRI and rETRII values than the vertical leaves (Figure 5A,B). Therefore, the horizontal leaves had stronger photosynthetic light use efficiency than the vertical leaves. Under low light, the horizontal and vertical leaves displayed similar low levels of the rETRI/rETRII ratio (Figure 5C). After transitioning to high light, the rETRI/rETRII ratio first increased to the peak in 10 s and then rapidly decreased over 30 s (Figure 5C). These results indicated the transient stimulation of CEF under FL, because the increased rETRI/rETRII ratio is an indicator of CEF activation.

PSI photoinhibition can be indicated by the decrease in the maximum photo-oxidizable P700 (*P_m_*) and the increase in Y(NA). After 24 min of FL treatment, the stable value of Y(NA) under high light increased more in the vertical leaves than in the horizontal leaves (Figure 6A). Furthermore, the vertical leaves showed significantly larger decreases in *P**_m_* than the horizontal leaves (Figure 6B). Further experiments indicated that FL-induced PSI photoinhibition was positively correlated to the average Y(NA) after transitioning to high light for 10 s (Y(NA)_10s_) (Figure 7A), indicating that the PSI overreduction was the main cause of FL-induced PSI photoinhibition. Interestingly, a tight relationship existed between the leaf angle and the extent of FL-induced PSI photoinhibition (Figure 7B), indicating that PSI was tolerant to FL in the horizontal leaves but was susceptible in the vertical leaves.

## 4. Discussion

In nature, leaves usually experience FL conditions [1,2,51]. During FL, a sudden increase in irradiance is accompanied with an immediate increased PSII electron transport and a slower kinetics of CO_2_ fixation [3,52]. Consequently, electrons transferred to PSI cannot be immediately consumed by electron sinks downstream [5,31]. The resulting PSI overreduction can cause PSI photoinhibition in *Arabidopsis thaliana* [6,53], rice [3], tobacco [37,38], and *Bletilla striata* [12,46]. Once the PSI activity is damaged, plant growth would be impaired, owing to the suppression of the photosynthetic electron flow and photoprotection [16,19]. CEF is critical for normal photosynthesis in angiosperms grown under FL [6,9,53]. In addition, the water–water cycle has the ability to rapidly consume excess light energy in PSI and thus avoid FL-induced PSI overreduction [39,54,55,56]. The downregulation of PSII activity or PSII electron flow can also alleviate FL-induced PSI photoinhibition [20,22,57,58]. A recent study suggested that FL-induced PSI photoinhibition is determined by the intensity difference between low and high light [46]. Although the leaf angle can affect the light absorption and thus influence the intensity of FL, no study has already investigated the role of leaf angle regulation in photoprotection under FL. An interesting observation is that mature leaves tend to be horizontal, but young leaves tend to be vertical in tobacco plants. However, the ecophysiological significance of this leaf angle difference has been rarely understood.

In this study, we documented that vertical young leaves have relatively lower photosynthetic light use efficiency than horizontal leaves (Figure 5). Under constant high light, most excess light energy in PSII can be dissipated harmlessly through nonphotochemical quenching, leading to the low level of Y(NO) (Figure 2). Concomitantly, PSI is oxidized, and PSI overreduction is prevented (Figure 2). Therefore, photoprotection for PSI and PSII are well-performed in vertical and horizontal leaves. However, FL causes stronger PSI photoinhibition in vertical leaves than in horizontal leaves (Figure 6 and Figure 7). In contrast, the response of PSII to FL does not differ between vertical and horizontal leaves (Figure 4). These results indicate that vertical young leaves are more sensitive to PSI photoinhibition when exposed to natural FL conditions. The vertical leaf angle in young leaves decreases the light absorption and thus diminishes the intensity of FL, i.e., intensity difference between low and high light, alleviating the PSI photoinhibition induced by FL. Therefore, the vertical leaf angle in young leaves plays a significant role in PSI photoprotection under FL. By comparison, the leaf angle in horizontal leaves increases the light absorption to maximize photosynthesis. Under such conditions, leaves will be exposed to a relatively stronger FL. Fortunately, horizontal leaves have the ability to protect PSI against FL. Therefore, the regulation of the leaf angle is an important strategy to maximize the photosynthetic light use efficiency and to strengthen photoprotection.

Previous studies have investigated the mechanism of PSI photoinhibition under FL [9,31,59]. In model plants, PSI photoinhibition is caused by oxidative damage induced by ROS generated within PSI [10,22,60]. Under anaerobic conditions or if PSI is oxidized, PSI photoinhibition hardly occurs [6,10]. Therefore, the existence of O_2_ and PSI overreduction is based on two prerequisites of PSI photoinhibition. At present, the transient PSI overreduction after transitioning to high light is thought to be the main cause of FL-induced PSI photoinhibition [46,61]. Consistently, our present study also indicates that transient PSI overreduction is the primary cause of FL-induced PSI photoinhibition (Figure 7).

During FL treatment, PSI overreduction was accelerated in vertical leaves when compared with horizontal leaves (Figure 3). The PSI redox state in photosynthetic organisms is adjusted through donor- and acceptor-side regulation. In donor-side regulation, a high ∆pH restricts the electron flow to PSI through downregulating plastoquinone oxidation at the cytochrome b6f complex [26,40,62]. After transitioning to high light, vertical and horizontal leaves showed a rapid induction of NPQ (Figure 4). This result suggests that the formation of ∆pH under FL did not differ between them, because the induction of thermal energy dissipation is largely dependent on ∆pH formation. Therefore, the stronger PSI overreduction in vertical leaves than horizontal leaves is not caused by the donor-side regulation of PSI. In the other regulatory mechanism (acceptor-side regulation of PSI), the electron sink downstream of PSI consumes the excess electrons and thus alleviates PSI overreduction. This regulation involves CO_2_ assimilation [61,63] and photorespiration (data not shown), the operation of O_2_ photoreduction mediated by flavodiiron proteins [30,31,34,64] and the water–water cycle [11,54,65]. Flavodiiron proteins do not exist in angiosperms [28], and transient PSI overreduction indicates that the activity of the water–water cycle in tobacco is negligible [54]. Therefore, the difference in the PSI redox state under FL between vertical and horizontal leaves is attributed to CO_2_ assimilation and photorespiration. Compared with vertical leaves, horizontal leaves have a higher Rubisco content, owing to the higher rETRI and rETRII values under high light (Figure 5). An increased Rubisco content can enhance photorespiration and thus consume excess electrons in the PSI. Therefore, the stronger PSI overreduction under FL in vertical leaves is likely linked to their relatively lower photosynthetic capacity.

## 5. Conclusions

Many previous studies have documented the molecular mechanisms of photoprotection for PSI against FL conditions. In this study, we, for the first time, point out the role of leaf angle regulation in photoprotection for PSI under FL. Upon a sudden transition from low to high light, vertical young leaves display stronger PSI overreduction than horizontal mature leaves. Consequently, after the same FL treatment, PSI photoinhibition is stronger in vertical young leaves than horizontal leaves. The vertical leaf angles of young leaves decrease the light absorption and thus alleviate PSI photoinhibition when exposed to natural FL conditions to avoid FL-induced PSI photoinhibition.

## Figures and Tables

**Figure 1 cells-11-00252-f001:**
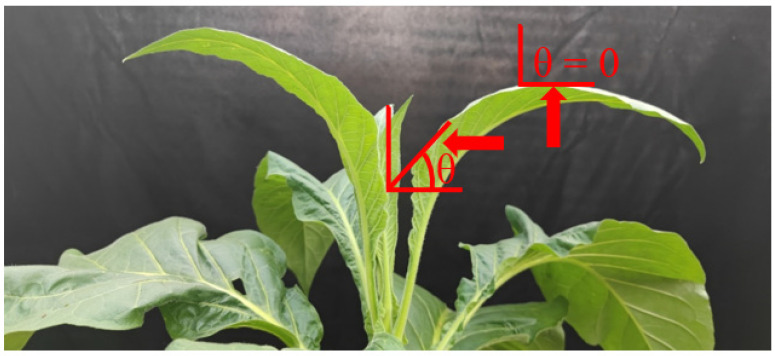
Photographs of tobacco leaves. The symbol θ indicates the leaf angle.

**Figure 2 cells-11-00252-f002:**
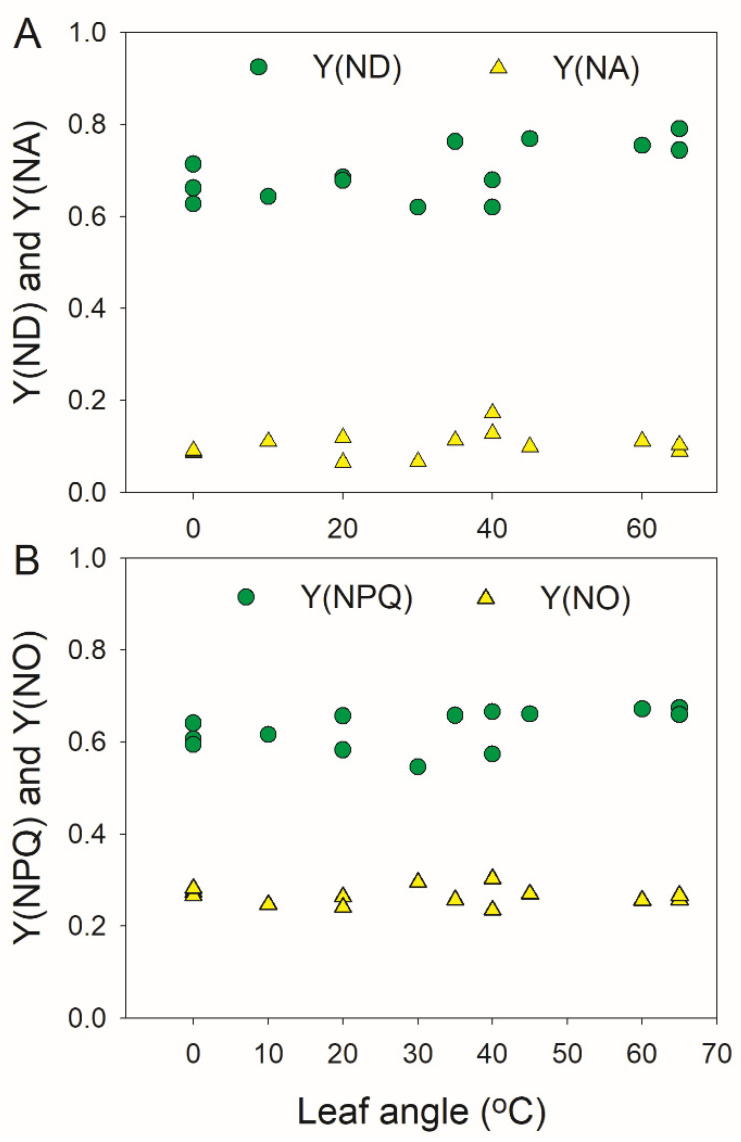
The relationships between the leaf angle and stable PSI (**A**) and PSII (**B**) parameters. After illumination at 1809-μmol photons m^−2^ s^−1^ for 20 min, the steady-state PSI and PSII parameters were recorded. Y(ND), the quantum yield of PSI nonphotochemical energy dissipation due to donor-side limitation; Y(NA), the quantum yield of PSI nonphotochemical energy dissipation due to acceptor-side limitation; Y(NPQ), the quantum yield of regulatory energy dissipation in PSII; and Y(NO), the quantum yield of nonregulatory energy dissipation in PSII.

**Figure 3 cells-11-00252-f003:**
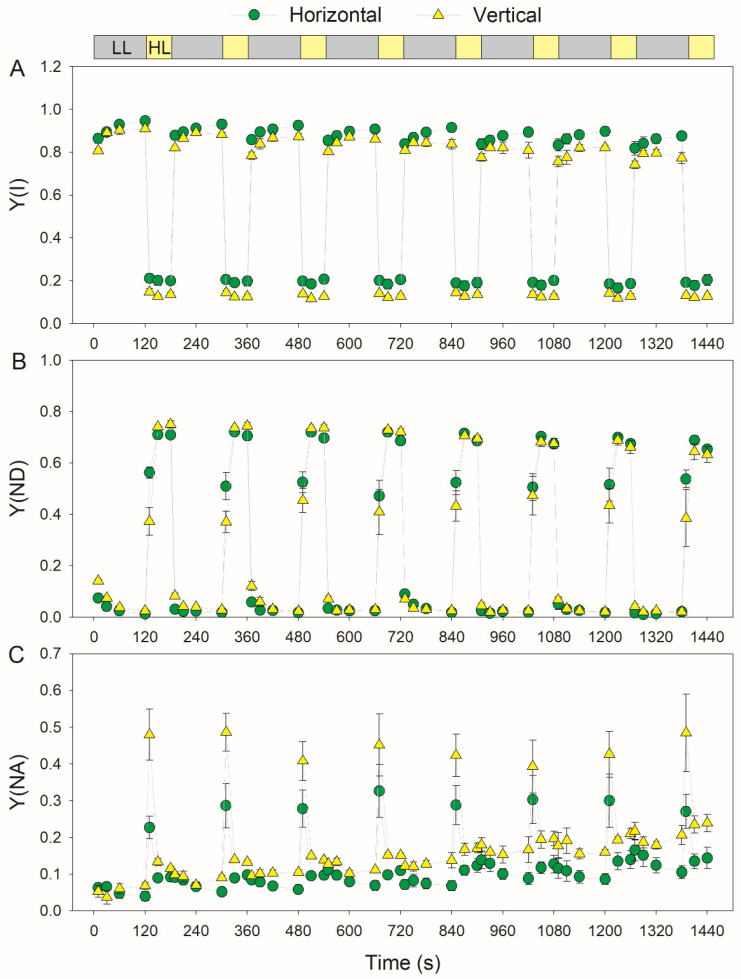
Changes in the PSI parameters under fluctuating light in the horizontal (leaf angle = 0°) and vertical (leaf angle = 60–65°) leaves. After illumination at 1809-μmol photons m^−2^ s^−1^ for 20 min for photosynthetic induction, the leaves were exposed to fluctuating light alternating between 59- and 1809-μmol photons m^−2^ s^−1^ every 2 min for 24 min. (**A**) Y(I), the quantum yield of PSI photochemistry; (**B**) Y(ND), the quantum yield of PSI nonphotochemical energy dissipation due to donor-side limitation; and (**C**) Y(NA), the quantum yield of PSI nonphotochemical energy dissipation due to acceptor-side limitation. Data are shown as the means ± SE (*n* = 3).

**Figure 4 cells-11-00252-f004:**
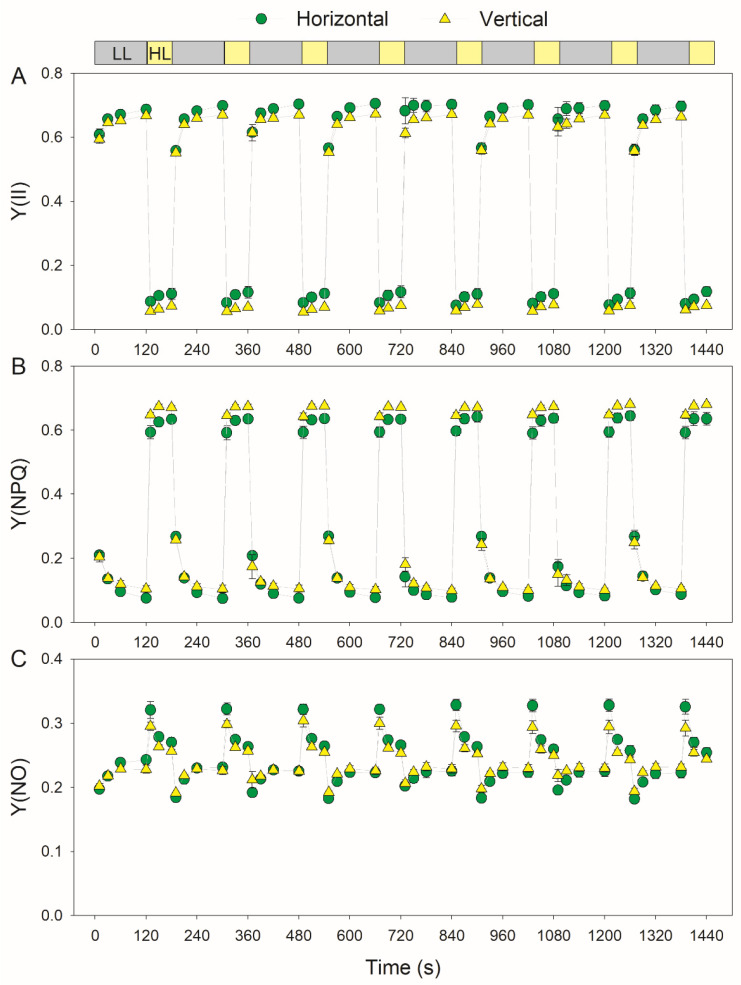
Changes in the PSII parameters under fluctuating light in the horizontal (leaf angle = 0°) and vertical (leaf angle = 60–65°) leaves. After illumination at 1809-μmol photons m^−2^ s^−1^ for 20 min for photosynthetic induction, the leaves were exposed to fluctuating light alternating between 59- and 1809-μmol photons m^−2^ s^−1^ every 2 min for 24 min. (**A**) Y(II), the effective quantum yield of PSII photochemistry; (**B**) Y(NPQ), the quantum yield of regulatory energy dissipation in PSII; and (**C**) Y(NO), the quantum yield of nonregulatory energy dissipation in PSII. Data are shown as the means ± SE (*n* = 3).

**Figure 5 cells-11-00252-f005:**
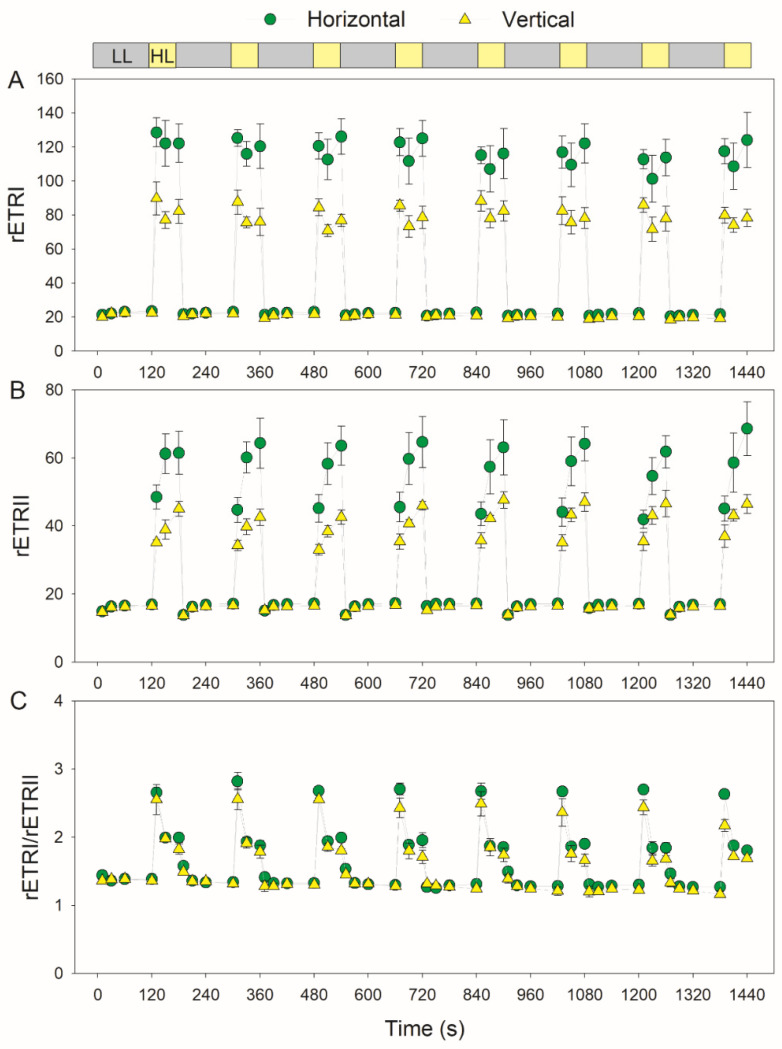
Changes in the relative electron transport rates under fluctuating light in the horizontal (leaf angle = 0°) and vertical (leaf angle = 60–65°) leaves. After illumination at 1809-μmol photons m^−2^ s^−1^ for 20 min for photosynthetic induction, the leaves were exposed to fluctuating light alternating between 59- and 1809-μmol photons m^−2^ s^−1^ every 2 min for 24 min. (**A**) rETRI, relative electron transport rate through PSI; (**B**) rETRII, relative electron transport rate through PSII; and (**C**) rETRI/rETRII, relative activation of cyclic electron flow. Data are shown as the means ± SE (*n* = 3).

**Figure 6 cells-11-00252-f006:**
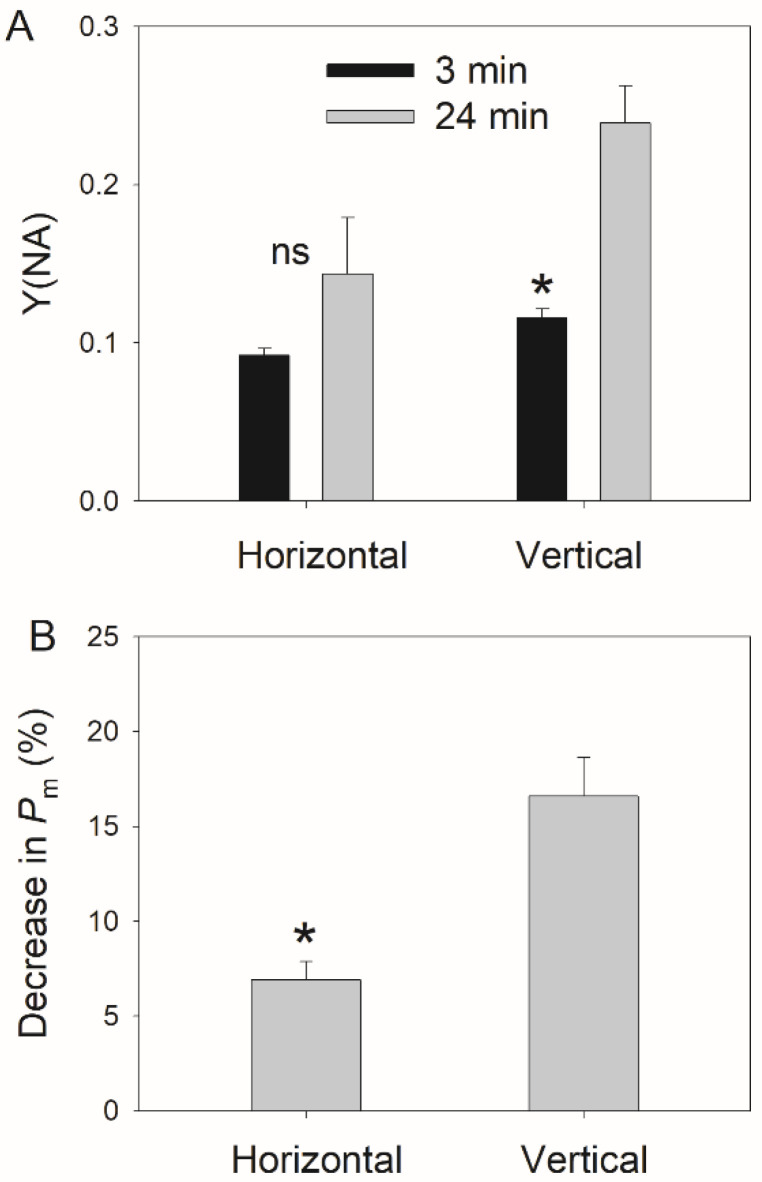
The effects of fluctuating light treatment on Y(NA) (**A**) and PSI activity (**B**) in the horizontal (leaf angle = 0°) and vertical (leaf angle = 60–65°) leaves. Asterisk indicates a significant difference between different treatments, and ns indicates no significant difference.

**Figure 7 cells-11-00252-f007:**
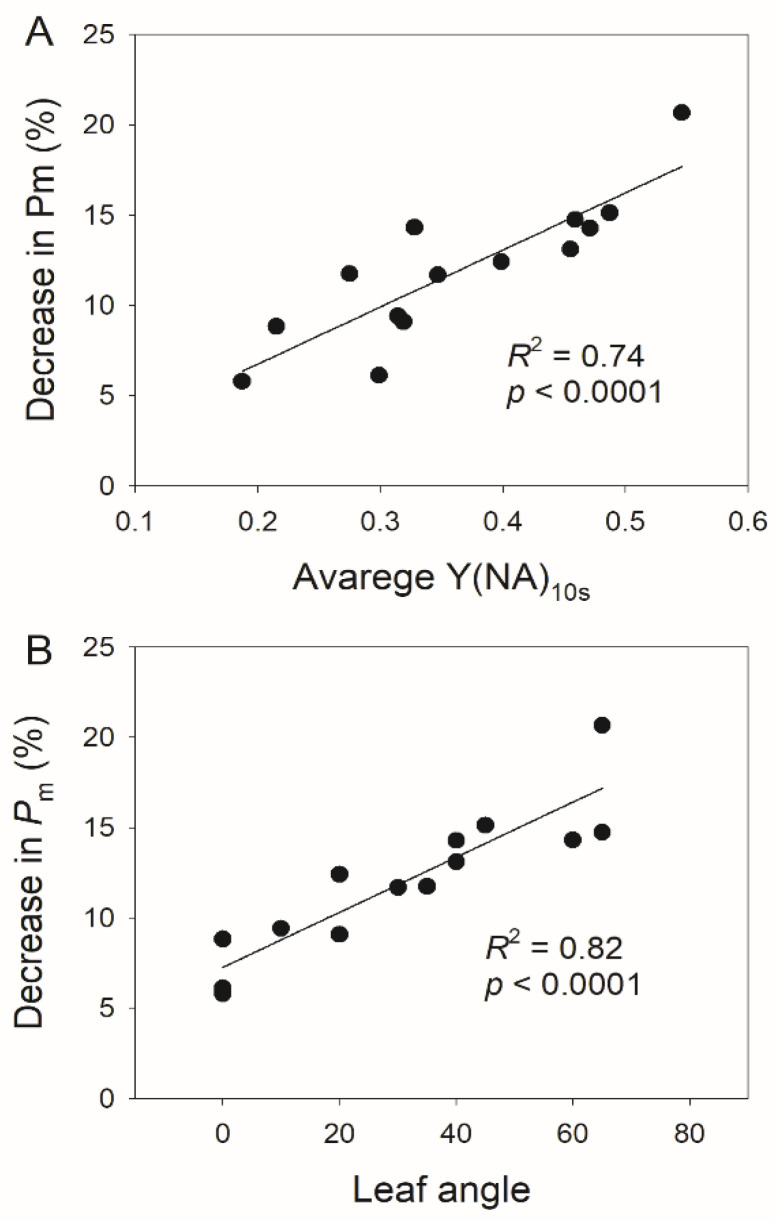
Relationships between FL-induced PSI photoinhibition and PSI overreduction (**A**) and the leaf angle (**B**). Y(NA)_10s_ indicates the average value of Y(NA) after the transition from low to high light for 10 s.

## Data Availability

The data presented in this study are available on request from the corresponding author.

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
