# Peer review of "Regulation of Leaf Angle Protects Photosystem I under Fluctuating Light in Tobacco Young Leaves"

_cells, 2022, doi:10.3390/cells11020252_

Round 1

Reviewer 1 Report

The authors analyzed relationship between the leaf angle and the harmful effects of fluctuating light on function of photosystem I in tobacco leaves. It was convincingly demonstrated that the leaf angle was tightly correlated to the extent of PSI photoinhibition induced by fluctuating light. Vertical young leaves were more susceptible to PSI photoinhibition than horizontal leaves when exposed to the same fluctuating light. Thus, the regulation of leaf angle is suggested as an important strategy used by young leaves to protect PSI against photoinhibition under fluctuating light. 

The study is very well introduced and the aims are clearly stated. The experimental design is appropriate and the description of methods is exhausting. The results are presented clearly and the conclusions are very well supported by the results. The discussion is good. 

Altogether, I consider this work as good and meaningful. It is very precisely elaborated and I have no specific comments. I recommend accepting the manuscript in present form. 

Author Response

Thanks a lot for the reviewer's positive evaluation.

Reviewer 2 Report

Avoid double superlatives e.g. Abstract and in the text: upon a sudden increase in illumination, PSI was highly over-reduced in ex-treme young leaves but was highly oxidized in mature leaves. 

M&M

Page 2 Line 92 two 2 days and were

Are these experiments done in one day? How the authors managed the leaf angle variation during the experiment in the day?

What was the time of the day, when the experiments were conducted.  

Page 3 Line 99 Subsequently, the PSI activity was measured three times for each leaf to decrease the error. At the same point or variable portions of the leaf. This may be explained and expanded, so that readers can repeat the experiments following your protocol.

The leaves were of the same plant or with similar leaf angle from different plants?

Are the readings done in detached leaves or attached leaves?

Results

Page 4 line 133: finely dissipated harmlessly via thermal energy dissipation, diminishing

Conclusion

‘Page 11 Line 278-281 : To avoid 277 FL-induced PSI photoinhibition, the leaf angle of young leaves is tended to be vertical. Such characteristics decrease light absorption and thus reduce the intensity difference between low and high light. Therefore, the vertical leaf angle is critical for photoprotection for PSI in young leaves exposed to natural field FL conditions.’

This conclusion seems too strong, as the experiment was done on both young and old leaves. There was no experimental design with only young leaves with same chronological age with different leaf angles to conclude this. The statement may be modified or deleted from the conclusion. It may be mentioned in the results or discussion section as a probability. It cannot be conclusive.

Author Response

Thanks a lot for these important comments.

  1. Double superlatives have been removed.
  2. This mistake has been revised.
  3. These experiments were conducted in different clear days. Before photosynthetic measurements, leaf angles were measured.
  4. We conducted these experiments in the morning (a.m. 9:00-11:00).
  5. Because Pm value might change slightly among independent measurements, Pm value was measured three times at the same point and the average Pm was calculated to decrease the error.
  6. Leaves with similar leaf angle from different plants were measured.
  7. All measurements were conducted in attached leaves.
  8. This sentence has been revised.
  9. The conclusion has been modified. Upon a sudden transition from low to high light, vertical young leaves display stronger PSI over-reduction than horizontal mature leaves. Consequently, after the same FL treatment, PSI photoinhibition is stronger in vertical young leaves than horizontal leaves. The vertical leaf angles of young leaves decrease light absorption and thus alleviate PSI photoinhibition when exposed to natural FL conditions.